# A Distributed Double-Loop Optimization Method with Fast Response for UAV Swarm Scheduling

**Runfeng Chen, Jie Li \*, Yiting Chen and Yuchong Huang**

College of Intelligence Science and Technology, National University of Defense Technology, Changsha 410073, China
* Correspondence: lijie09@nudt.edu.cn

**Abstract:** An unmanned aerial vehicle (UAV) swarm has broad application prospects, in which scheduling is one of the key technologies determining the completion of tasks. A market-based approach is an effective way to schedule UAVs distributively and quickly, meeting the real-time requirements of swarm scheduling without a centre. In this paper, a double-loop framework is designed to enhance the performance of scheduling, where a new task removal method in the outer loop and a local redundant auction method in the inner loop are proposed to improve the optimization of scheduling and reduce iterations. Furthermore, a deadlock detection mechanism is introduced to avoid endless loops and the scheduling with the lowest local cost will be adopted to exit the cycle. Extensive Monte Carlo experiments show that the iterations required by the proposed method are less than the two representative algorithms consensus-based bundle algorithm (CBBA) and performance impact (PI) algorithm, and the number of allocated tasks is increased. In addition, through the deadlock avoidance mechanism, PI can completely converge as the method in this paper.

**Keywords:** UAV swarm; distributed scheduling; fast response; market-based approach; double-loop optimization

## 1. Introduction

In the past few years, unmanned aerial vehicle (UAV) swarms have been booming, which have great application prospects in search and rescue, logistics distribution, environmental monitoring, and so on [1–4]. In these applications, scheduling is one of the key technologies that determines whether the UAVs can complete the tasks and how well they complete the tasks. Because the situation, such as UAVs, and tasks change in real time, and the tasks have spatio-temporal constraints, this brings a non-trivial challenge to the real-time and optimization of scheduling [5,6].

The multi-UAV scheduling problem is a variant of the travelling salesman problem (TSP), which is a well-known NP-hard combinatorial optimization problem in graph theory [7,8]. It has more complicated constraints, such as a time constraint that requires the task to be completed by a certain deadline, while the execution time of tasks is coupled with the UAV's path. In addition, there are more UAVs needed to schedule, which requires more computation and exacerbates the original combinatorial explosion problem [9,10].

The centralized method is a traditional way to solve the above scheduling problem, where the UAV, task and situation information are gathered in a centralized computing node and it calculates the schedules and distributes them to the UAVs. If the information needed for the centralized computing node is comprehensive and accurate, the optimal scheduling can be calculated by this method, but the computation time increases dramatically with the increase in the number of UAVs and tasks [11,12]. Even using heuristic algorithms, such as the genetic algorithm [13,14], ant colony algorithm [15,16] and particle swarm optimization [17,18], it is difficult to meet the timeliness of swarm scheduling when the scale is not small. In addition, a large amount of communication is required to obtain the

situation and other information required by the centralized computing node, which brings great pressure on communication [19–21]. Moreover, once the centralized computing node is damaged, the swarm will collapse completely, making it very vulnerable [22,23].

The distributed method is a centreless computing method, where every UAV is a computing node and conflict-free scheduling can be obtained through the interaction for consensus. In this way, if a UAV fails to complete the task or crashes, triggering rescheduling, it can receive a new schedule from the other UAVs to complete the unfinished task, this method has strong flexibility and robustness. Furthermore, through the individual greedy strategy and the mechanism that each individual only calculates its schedule, the calculation amount can be greatly reduced and the scheduling can be obtained quickly, which allows for a real-time response to rapidly changing environments or emerging tasks [24,25]. Although the optimization performance of the distributed method is not as good as the centralized method, its low communication and low computation required by the airborne platform, its flexibility, rapidity, and robustness also required by the UAV swarm, and the scheduling scheme has good performance [26,27].

The market-based approach is the most competitive distributed method because it is naturally consistent with the characteristics of the swarm. The most representative scheduling algorithms, the consensus-based bundle algorithm (CBBA) [26] and performance impact (PI) algorithm [28] and their extensions [29–31] are market-based methods. Through several iterations of the 'scheduling calculation and conflict resolution' two stages, the swarm scheduling without task conflict is obtained. However, since communication is limited, too many iterations will increase the communication delay, and then in turn increase the iterations to resolve task conflicts. Therefore, reducing the number of iterations to reduce the dependence on communication affected by the environment is one directions of this research. In addition, due to the lack of global knowledge, some time-sensitive tasks may not be completed. Therefore, improving the optimality to increase the number of allocated tasks is the second research direction of this paper.

Among so many market-based methods, the PI algorithm is a state-of-the-art method, which updates the impact of the task on the current schedule in real time, truly reflecting the cost of the task [28,30,31]. However, updating the task cost in real time also has some drawbacks, such as more iterations and occasional deadlock problems, which generates a algorithm with low iterations and even infinite iterations without convergence. More iterations mean more time to generate a schedule, and falling into infinite iterations means that the algorithm can never give the scheduling results, greatly affecting the reliability and performance of the scheduling algorithm. In addition, the way the PI algorithm removes tasks outbidden by other UAVs also limits its exploration of tasks, so the optimality can be further improved.

To reduce the iterations required for scheduling to improve the real-time performance, while avoiding deadlock to improve reliability and increase task exploration to improve optimization of the algorithm, this paper performs the following work:

(1)　A new double-loop framework is designed, where the outer loop is an information consensus and the inner loop is a local optimization. The former is that UAVs communicate to obtain consistent scheduling and remove tasks outbidden by other UAVs; the latter is to select the optimal task based on the consistent scheduling obtained by the outer loop.

(2)　A new task removal method is proposed, which has better task exploration and ensures the consensus of scheduling in the outer loop of this iteration. The range of tasks to choose is increased and the conflict of tasks in the outer loop is reduced, which is helpful to improve the optimization of scheduling and reduce the iterations required by the algorithm.

(3)　A local redundant auction method is proposed to reduce the iterations further, where the UAV only considers other UAVs related to the task but not all UAVs, eliminates potential task conflicts in advance and avoids consuming computing power for irrelevant UAVs, balancing calculation time and iterations to quickly obtain schedules.

(4)  A deadlock detection mechanism is introduced to detect the endless loop and avoid it by recording and comparing local scheduling for each iteration. Once the same local scheduling is found, the UAV will be judged to enter the deadlock and the scheduling with the lowest local cost will be adopted to exit the cycle.

(5)  Extensive Monte Carlo experiments show that the iterations required by the proposed method are less than the two representative algorithms CBBA and PI, which are reduced by about 40 and 55%, respectively. Furthermore, the number of allocated tasks is increased by 4–8% and 3–7% respectively, and the improvement ratio is 26–90% and 15–54% respectively. Through the deadlock avoidance mechanism, PI can completely converge as the method in this paper.

The rest of the paper is organized as follows. Section 2 formulates the multi-UAV scheduling problem and introduces solutions. Section 3 introduces the idea and framework of the proposed method, as well as the details of the method. Furthermore, a large number of experiments are shown in Section 4 to verify the validity of the proposed framework and method. Finally, Section 5 concludes this paper.

## 2. Preliminaries

### 2.1. Problem Formulation

Multi-UAV scheduling is where $n$ UAVs allocate $m$ tasks to form their own schedules, whose goal is to minimize the total cost of completing all tasks by UAVs. Each UAV $i$'s schedule $\mathbf{p}_i$ is the execution sequence and time of allocated tasks, satisfying constraints of tasks such as starting time and deadline. Generally, it could be formulated as a constrained optimization problem with objectives formulated as follows:

$$\min \sum_{i=1}^{n} \left( \sum_{j=1}^{m} c_{ij}(\mathbf{p}_i) x_{ij} \right) \tag{1}$$

subject to:

$$\sum_{i=1}^{n} x_{ij} \leq 1 \quad \forall j \in \mathcal{J} \tag{2}$$

$$\sum_{j=1}^{m} x_{ij} \leq m \quad \forall i \in \mathcal{I} \tag{3}$$

$$t_{ij} \leq d_j \quad \forall i \in \mathcal{I}, \quad \forall j \in \mathcal{J} \tag{4}$$

where $x_{ij}$ is a binary variable, when $x_{ij} = 1$, UAV $i$ performs task $j$, otherwise the vice versa. $c_{ij}(\mathbf{p}_i)$ is the cost for UAV $i$ to perform task $j$ according to the schedule $\mathbf{p}_i$, the estimated time for UAV $i$ to perform task $j$ is usually adopted. UAV scheduling also needs to consider its own and task constraints, for example, the first constraint indicates any task $j$ can only be performed by one UAV at most; the second indicates UAV $i$ can perform $m$ tasks at most; the third is the time $t_{ij}$ of any task $j$ performed by any UAV $i$ cannot exceed task $j$'s deadline $d_j$. There may be other similar constraints such as UAV endurance, processing capacity, load capacity, etc., which are not presented here.

### 2.2. Solving Approach

The market-based approach is an effective way to solve the above problems distributively and quickly, which mainly includes two stages: scheduling calculation and conflict resolution, as shown in Figure 1. In the scheduling calculation stage, each UAV calculates its optimal tasks with bids to form its schedules based on its local information, where each task in the UAV schedules must satisfy the constraints such as task deadline, UAV endurance, etc. Then, in the conflict resolution stage, each UAV resolves task conflicts with other UAVs by communicating schedules, where each UAV $i$ generally communicates three vectors: task winners $\mathbf{z}_i$, task-winning bids $\mathbf{y}_i$ and timestamps $\mathbf{t}_i$. Through two-stage iteration, conflict-free scheduling is obtained.

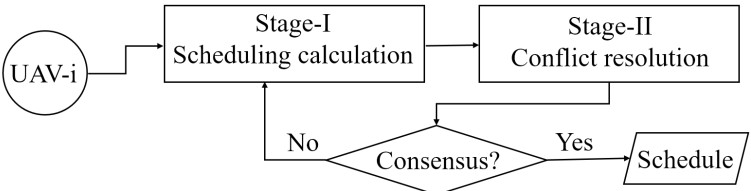

**Figure 1.** The general flow of the market-based approach.

PI is the most optimized method among the current market-based methods because it updates the impact of the task on the current schedule in real time, which truly reflects the cost of the task [28,30,31]. However, updating the task cost in real time also brings the disadvantage of a large number of iterations required, and occasionally causes deadlock problems, which makes the algorithm fall into an infinite loop. In addition, the way it removes tasks outbidden by other UAVs also limits its exploration of tasks, so the optimality can be further improved. In this way, since the bid of PI is the impact of a task on a schedule, smaller is better.

## 3. Method

A distributed double-loop optimization method is proposed in this section, where Section 3.1 introduces its idea and framework and Sections 3.2 and 3.3 introduce the outer-loop consensus and inner-loop optimization, respectively. In Section 3.4 a endless loop avoidance mechanism is proposed to ensure the convergence of the PI method and similar methods with real-time bid updates, where the optimal conflict-free scheduling will be obtained when exiting from the loop if deadlock exists. Finally, the overall flow chart of the proposed method and the computational complexity analysis are shown in Section 3.5.

### 3.1. Basic Idea and Framework

In the traditional auction framework, each UAV only considers its own scheduling, rather than the scheduling of other UAVs, which can greatly reduce the time required for calculation, but also increases the task conflict among UAVs, which requires multiple iterations to achieve consistent scheduling. Furthermore, in the traditional redundant framework, each UAV considers the scheduling of all UAVs, so consistent scheduling can be obtained as long as the information input obtained by each UAV is consistent. However, the computing amount of the UAV increases exponentially with the number of UAVs and tasks, requiring a long scheduling time. These two frameworks cannot meet the requirements of typical multi-UAV distributed scheduling such as UAV swarm. How to balance calculation time and iterations to quickly obtain scheduling is an important topic, which is also the focus of this paper.

To reduce the iterations, it is necessary to reduce task conflicts among UAVs. Each UAV cannot consider its own scheduling but also cannot consider the scheduling of all UAVs which brings a sharp increase in computing time. The core idea of the method in this paper is that each UAV only considers the UAVs that have potential task conflict with it to reduce the iterations and avoid the sharp increase in computation.

In this paper, a new double-loop framework is designed, where the outer loop is a consensus process just like the auction framework, and the inner loop is a local optimization process. In the outer loop, each UAV communicates with its neighbouring UAVs to resolve task conflicts among UAVs and obtain consistent global scheduling. Then each UAV removes tasks that conflict with consistent global scheduling from its individual scheduling so that all UAVs obtain consistent conflict-free scheduling. In the inner loop, each UAV tries to add new tasks to its individual scheduling based on the consistent scheduling obtained from the outer loop, hoping to adjust the global scheduling to a better level. There are two main steps in this process: local auction, where each UAV constructs its own internal auction process composed of UAVs with potential task conflicts, to resolve possible task conflicts with other UAVs in advance, reducing the iterations; deadlock detection,

through previous studies and experiments, it is found that although task bid updating has better optimization, it will occasionally deadlock, resulting in infinite iterations and non-convergence. This situation can be avoided through the endless loop avoidance method proposed in this paper. The double-loop framework designed in this paper is shown in Figure 2.

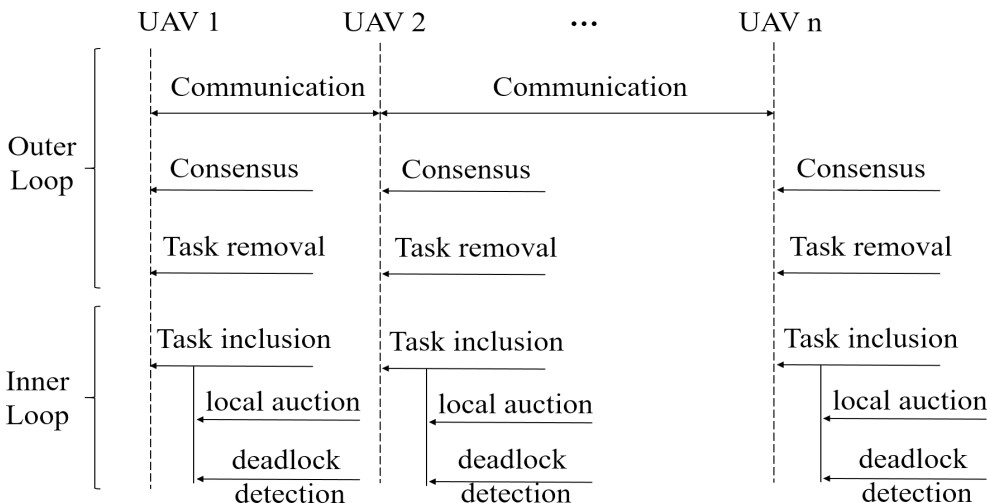

**Figure 2.** The double-loop framework designed in this paper.

### 3.2. Outer-Loop Consensus

The outer loop is an information consensus loop, where each UAV communicates with other UAVs, interacts with the local information about tasks among UAVs, uses the same consensus protocol to resolve the task conflicts between UAVs, and obtains consistent scheduling. Among them, individual tasks that conflict with consistent scheduling need to be removed according to removal rules.

Different from the PI removal rule, this paper adopts the rule of 'remove all tasks outbid by other UAV'. The formula is as follows:

$$\mathbf{p}_i' = \{j|j \in \mathbf{p}_i, j \notin \mathbf{d}_i\}, \quad \forall j \in \mathcal{J}, \quad \forall i \in \mathcal{I} \tag{5}$$

$$\mathbf{d}_i = \mathbf{p}_i[\mathbf{z}_i[\mathbf{p}_i] \neq i], \qquad \forall i \in \mathcal{I} \tag{6}$$

where the vector $\mathbf{p}_i'$ is the schedule after removing all outbid tasks, the vector $\mathbf{d}_i$ is the set composed of all tasks to be removed by UAV $i$, and the tasks to be removed are tasks outbid by other UAVs in the original scheduling $\mathbf{p}_i$ of UAV $i$, which is specifically reflected in that the winner of these task in the consistent task winner list $\mathbf{z}_i$ is not UAV $i$.

Figure 3 compares two different removal methods, in which PI is to retain outbid tasks if they can be retained, and the rule of removing the maximum positive cost difference is adopted. During each removal of a task, the removal performance impact (RPI, similar to the bid) of the task for the UAV is updated. This method helps preserve the selected task, but it also limits the selection of other tasks and may preserve the original task conflict. For example, UAV 1's task $T_2$ was defeated by UAV 2, but after it removed task $T_3$ that was also defeated by UAV 2, UAV 1's path changed, and it updated the RPI of task $T_2$, so that it outbid the current bid of UAV 2 and kept task $T_2$. The same operation also happened in UAV 2's task $T_4$. Although tasks $T_2$ and $T_4$ are retained in UAV 1 and UAV 2, respectively, the conflict of tasks $T_2$ and $T_4$ between UAV 1 and UAV 2 is also retained, and the exploration of other tasks is limited by the retention of tasks. However, the proposed method is to remove all outbid tasks. On the one hand, it ensures the consensus of all UAVs'

information; on the other hand, it enables the UAVs to explore more potential and better task choices.

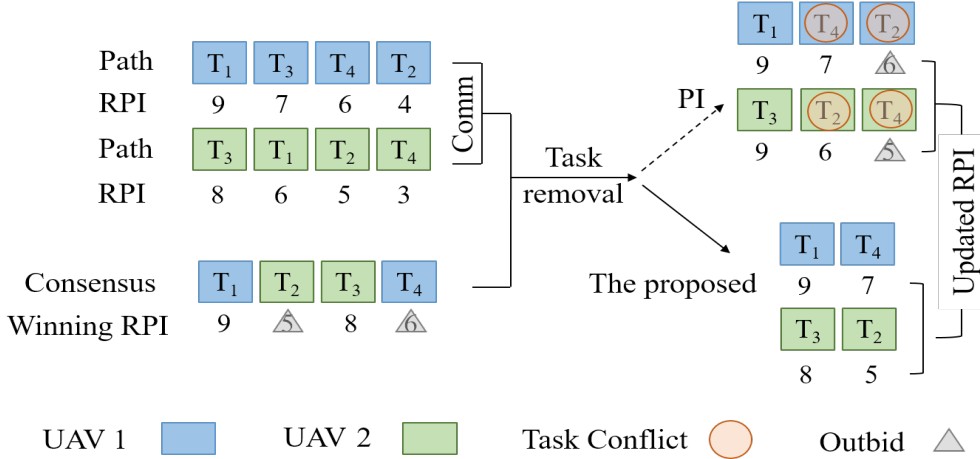

**Figure 3.** Comparison of two different removal methods.

The main process of the outer loop is shown in Algorithm 1. First, UAV $i$ communicates with all the UAVs $k$ in its neighbourhood to resolve task conflicts (line 1). Based on the communication content, the UAV set $\mathbf{L}_i$ related to UAV $i$ is obtained to prepare for the inner loop optimization (line 2). Second, UAV $i$ removes all tasks outbid by other UAVs and updates its scheduling time (lines 3–9). Finally, UAV $i$ updates its current scheduled bid and winner bid (line 10).

---

**Algorithm 1** Outer loop.

---

1: UAV $i$ communicates its scheduling with all neighbour UAV $k$ and resolves task conflicts [26]
2: get task-related UAVs $\mathbf{L}_i$ (Equation (8)) for inner loop
3: **for each** task $j$ in UAV $i$'s schedule $\mathbf{p}_i$ **do**
4:     **if** conflict: $\mathbf{z}_i[j] \neq i$ **then**
5:         remove task $j$ from schedule $\mathbf{p}_i$
6:         reset the winner and winnerbids of task $j$
7:     **end if**
8:     update the time of current schedule
9: **end for**
10: update the bid and winnerbids of current schedule

---

### 3.3. Inter-Loop Optimization

The inner loop is a local optimization, where each UAV independently selects the optimal task to include in its own scheduling, based on the consistent scheduling obtained by the outer loop. Among them, the task inclusion rules of UAVs determine the conflict and optimization of scheduling. This paper focuses more on scheduling conflict and expects to reduce the iterations and scheduling time by reducing task conflict among UAVs.

This paper proposes a local redundant auction method, in which the UAV only considers other UAVs related to the task, eliminates potential task conflicts in advance and avoids consuming computing power for irrelevant UAVs. The formula is as follows:

$$\mathbf{p}_i^{''} = a_i\left(\mathbf{p}_i^{'}, \mathbf{p}_k^{'}, \mathbf{q}_j\right), \quad \forall j \in \mathcal{J}, \quad \forall k \in \mathbf{L}_i, \quad \forall i \in \mathcal{I} \tag{7}$$

where $\mathbf{p}_i^{''}$ is the estimated scheduling of UAV $i$ by the constructed local auction $a_i$ based on the scheduling of UAV $i$ and UAV $k$ in the consensus of the last outer loop and the known

information of task $j$. $\mathbf{L}_i$ is the set of all UAVs that have task conflict with UAV $i$, defined as follows:

$$\mathbf{L}_i = \mathbf{z}_k[\mathbf{z}_k[\mathbf{p}_i] \neq \mathbf{z}_i[\mathbf{p}_i]], \quad \forall i, k \in \mathcal{I} \tag{8}$$

Figure 4 is the schematic diagram of inner-loop optimization. It can be seen that UAV 2 will not calculate for UAV 1, because there is no task conflict between them. Therefore, the local auction constructed by UAV 2 only contains UAV 1 related to tasks 2–4.

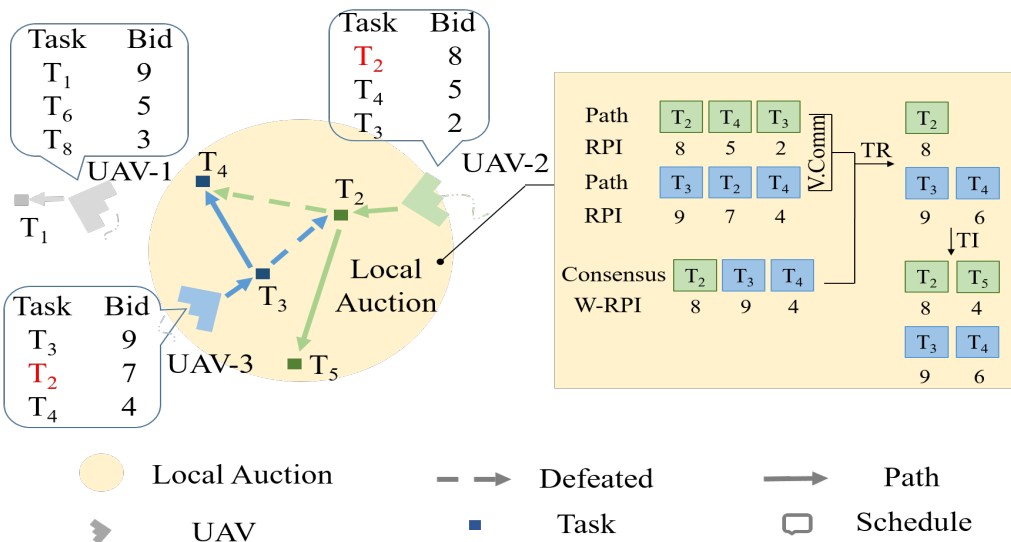

**Figure 4.** Schematic diagram of inner-loop optimization, where V.Comm is virtual communication, W-RPI is winning removal performance impact, TR is task removal, TI is task inclusion.

The main process of the inner loop is shown in Algorithm 2. If the UAV set $\mathbf{L}_i$ associated with UAV $i$ is not empty, a virtual local auction is constructed to estimate the optimal scheduling $\mathbf{p}_i$ of UAV $i$ (lines 1–3), and to detect whether deadlock exists and avoid it (line 4). If the set $\mathbf{L}_i$ is empty, UAV $i$ adds all feasible tasks that outbid all other UAVs (lines 5–7). Finally, UAV $i$ updates its winners list $\mathbf{z}_i$ and winner bid list $\mathbf{y}_i$.

---

**Algorithm 2** Inner loop.

1: **if** $\mathbf{L}_i \neq \varnothing$ **then**
2:     construct a virtual local auction $a_i \leftarrow i, \mathbf{L}_i$.
3:     estimate the best schedule $\mathbf{p}_i''$ by Formula (7).
4:     detect deadlock and avoid it if there is.
5: **else**
6:     UAV $i$ adds all the feasible task $j^*$ outbidding other UAVs to its schedule $\mathbf{p}_i''$.
7: **end if**
8: update UAV $i$'s winner $\mathbf{z}_i$ and winnerbids $\mathbf{y}_i$.

---

### 3.4. Endless Loop Avoidance

Similar to the PI method, the UAV will update the bid of the existing task after removing or including tasks each time, so as to obtain the real impact of the task on the scheduling of the UAV. However, this also brings the same deadlock problem as PI, as shown in Figure 5.

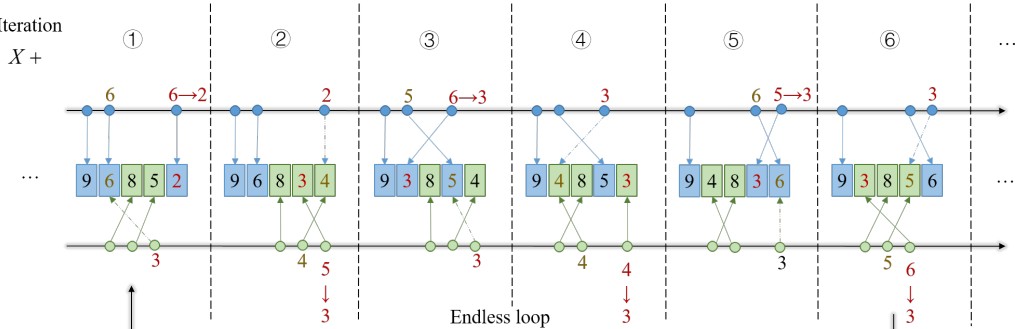

**Figure 5.** An example of a deadlock: a series of problems caused by updating the bid, where the number in the figure is the bid of the task.

In Figure 5, blue represents UAV 1 and green represents UAV 2. The five boxes in each iteration represent the consistent assignment of the five tasks after the communication, and the internal value is the winning bid of the task. It can be found that in the sixth iteration, the schedule of UAV 1 is [T1 T5 T4], and the corresponding bid is [9 6 3]; the schedule of UAV 2 is [T3 T4 T2], and the corresponding bid is [8 5 3]. UAV 1 lost the bid for task 4, but UAV 2's bid for task 2 was updated from 6 in the previous iteration to 3, so UAV 1 launched a bid for task 2 in the next iteration, bidding for 6, and it went back to the first iteration.

According to a large number of experimental statistics and analyses, in all cases, the reason why the UAVs fall into a dead cycle is that the UAVs repeatedly compete for tasks back and forth. Because the task bid is updated in this iteration and other UAVs acquire it in the next iteration, the delay in updating the task bid leads to the wavering of the UAVs between removing and adding tasks. In order to avoid such a dead cycle, a detection mechanism is proposed in this section. Based on the historical data of the local scheduling of the UAV, once the same local scheduling is found in the local auction of the inner loop, the UAV will be judged to enter the scheduling deadlock and the scheduling with the lowest local cost will be adopted to exit the cycle. The formula is as follows:

$$\mathbf{p}_i^* = \left\{ \mathbf{p}_i^{t^*} | t^* = \arg\min_t C_i^t \right\}, \quad \forall t \in \Gamma', \quad \forall i \in \mathcal{I} \tag{9}$$

where $C_i^t$ is the local cost of the $t$-th iteration, and $\Gamma'$ is the collection of the iteration index of the maximum allotments in the local auction, which are defined as follows:

$$C_i^t = \sum_{k=1}^{|\mathbf{L}_i|} c\left(\mathbf{p}_k^t\right), \quad \forall k \in \{i\} \cup \mathbf{L}_i, \quad \forall t \in \Gamma', \quad \forall i \in \mathcal{I} \tag{10}$$

$$\Gamma' = \arg\max_t \sum_{k=1}^{|\mathbf{L}_i|} \left|\mathbf{p}_k^t\right|, \quad \forall k \in \{i\} \cup \mathbf{L}_i, \quad \forall t \in \Gamma, \quad \forall i \in \mathcal{I} \tag{11}$$

where in the $t$-th iteration, the time cost and the number of allocated tasks corresponding to the scheduling $\mathbf{p}_k$ of UAV $k$ are $c\left(\mathbf{p}_k^t\right)$ and $\left|\mathbf{p}_k^t\right|$, respectively, and $\Gamma$ is the set of iteration indexes of the endless loop.

Figure 6 shows the process of the method in this section to solve the example of the dead loop in Figure 4. The UAV detects the dead loop through the historical data and then finds the iteration with the maximum number of tasks allocated and the lowest local cost in the iteration of the loop. The corresponding UAV scheduling in the iteration is the optimal scheduling.

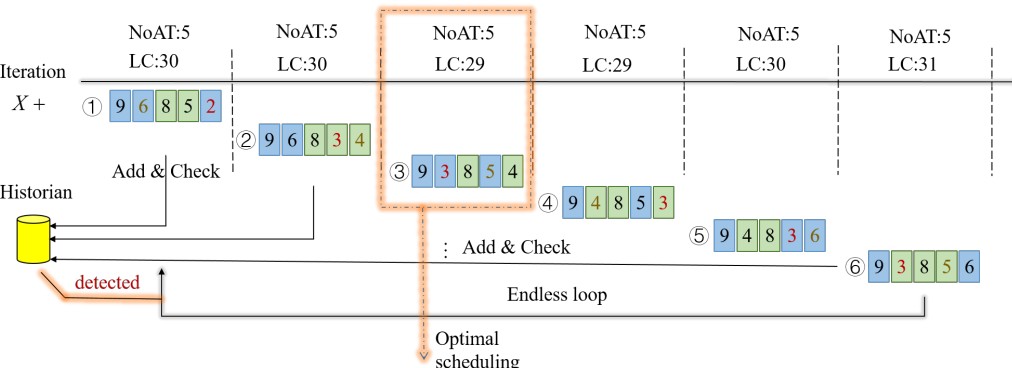

**Figure 6.** The process to solve the dead loop. NoAT is short for 'number of allocated tasks' and LC is short for 'local cost'.

Algorithm 3 summarizes the approximate process of endless-loop avoidance. Firstly, the historical data logger $\varpi$ and the completion flag $\omega$ are initialized in each inner-loop optimization (line 1). Then, in each iteration $t$, whether the obtained consistent scheduling $\mathbf{p}^t$ was in the historical data is checked (lines 4–9). If it was, the best scheduling from the historical data will be found and set the completion flag to exit the loop (lines 5–6). If not, the scheduling $\mathbf{p}^t$ is added to historical data (line 8).

---

**Algorithm 3** Endless-loop avoidance.

---

1: initializes historical data $\varpi = \varnothing$ and doneFlag $\omega = 0$
2: **while** $\omega = 0$ **do**
3:     get the consistent scheduling $\mathbf{p}^t$ of the current iteration $t$.
4:     **if** $\mathbf{p}^t \subset \varpi$ **then**
5:         find the best scheduling in the historical data by Formulas (10)–(12).
6:         set the doneFlag to break the loop: $\omega = 1$.
7:     **else**
8:         add the scheduling to historical data $\varpi \leftarrow \mathbf{p}^t$.
9:     **end**
10: **end while**

---

### 3.5. Global Analysis

The overall flow chart of the proposed method is shown in Figure 7, which mainly contains two loop iterations, outer loop and inner loop. In the outer loop, UAV $i$ obtains information about tasks from other UAVs through neighbouring communication, then resolves the task conflict among UAVs to obtain a consistent scheduling, and removes the conflicting tasks from its local schedule. Then it enters the inner loop, calculates the estimated conflict-free tasks that meet the constraints through local auction, and adds these tasks to its own schedule. Then it checks whether there is a deadlock, and finds the best conflict-free schedule in the cycle if there is a deadlock. If there is no infinite cycle, it judges whether a swarm consensus has been reached. If the UAV is inconsistent about tasks with other UAVs and there are still task conflicts, it will enter the outer loop again, and two loops will iterate until the inner-UAV consensus, and the conflict-free optimal schedule can be obtained.

The computational complexity of the method is mainly in the task inclusion, which is $O(k(m - |\mathbf{p}_i|)|\mathbf{p}_i|^2\vartheta_y\sigma)$, where $k$ is the controllable prediction number of UAVs, $\sigma$ is the calculation amount required to calculate the execution time of a task, and $\vartheta_y$ is the number of candidate tasks that satisfy constraints but have not be added to agent $i$'s schedule. The inclusion complexity of $O((m - |\mathbf{p}_i|)|\mathbf{p}_i|^2\vartheta_y\sigma)$ is the same as that of the method of CBBA and PI, which has been analysed in [28]. Of course, this is the theoretical maximum computational complexity, and the actual required amount of computation is much less, and the literature [27,32–34] also verifies the feasibility of real deployment.

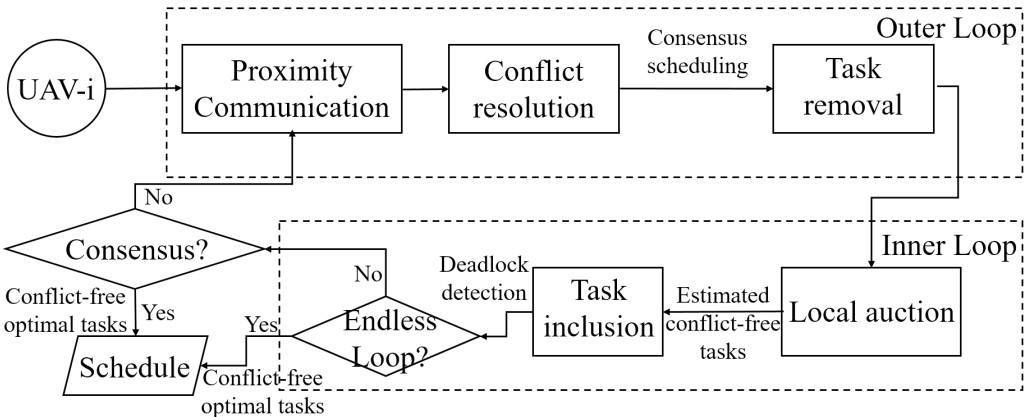

**Figure 7.** The overall flow chart of the proposed method.

## 4. Experiment

In order to verify the effectiveness of the proposed method, some comparisons are made with the two most representative algorithms, CBBA [26] and PI [28], where the scheduling results are compared with various numbers of randomly distributed UAVs and tasks. To make a more convenient and objective comparison, the test scenario adopted in this paper is the same as that in the PI method [28,30,31], where two types of UAVs in the swarm should go to complete two different types of tasks, and the specific parameters are shown in Table 1. Furthermore, for purpose of avoiding the error caused by small sample experiments, this paper uses a large number of Monte Carlo experiments, where each test with randomly distributed UAVs and tasks under the same $p$ and $n$ were performed 100 times. $n$ is the number of UAVs, $m$ is the number of tasks, and $p = m/n$ is the ratio of the number of tasks and the number of UAVs. If the UAV scheduling did not change three times, it was considered to converge.

**Table 1.** The settings of the simulation.

| Objects | Properties | Descriptions |
|---------|-----------|--------------|
| UAV | location | randomly distributed in 10 km × 10 km × 0 km area |
|  | speed | providing food: 50 m/s |
|  |  | providing medicine: 30 m/s |
| Task | location | randomly distributed in 10 km × 10 km × 1 km area |
|  | deadline | randomly distributed in [0, 2000 s] |
|  | duration | delivering food: 300 s |
|  |  | delivering medicine: 350 s |

Figure 8 shows the comparison of the three methods in terms of the number of iterations, and it can be found that the proposed double-loop optimization method (abbreviated as DL) has the least number of iterations, thus the scheduling results can be obtained faster. When $p = 3$ and $n = 6$, the median iterations of CBBA is 7, PI is 9 and the proposed method is 4, reduced by 43 and 56%, respectively. When $p = 5$ and $n = 16$, the iterations of CBBA are 10, PI is 14 and the proposed method is 6 with 40 and 57% reduction, respectively. This effectively verifies the effectiveness of the inner-loop optimization in the proposed double-loop architecture, and the local redundant auction can effectively resolve the potential conflicts with related UAVs.

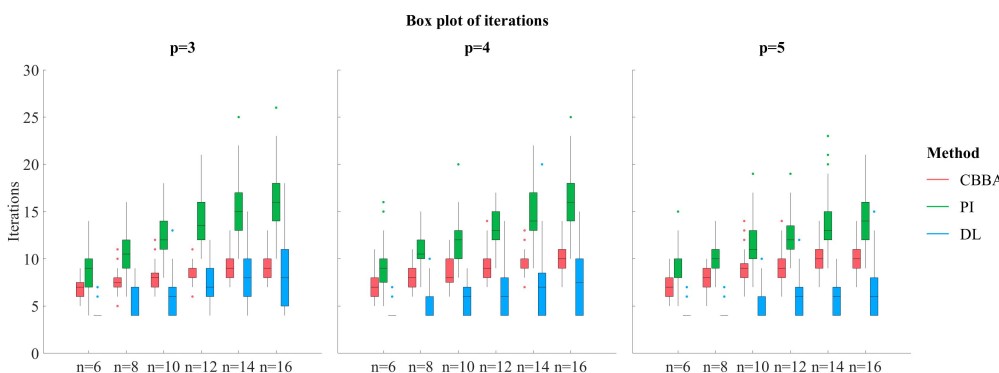

**Figure 8.** Comparison of three methods in terms of iterations.

Figures 9 and 10 show the proportion and increase in the improvement of the proposed method over the other two methods in the number of task allocations, respectively, where the grey part is the comparison between the proposed method and the PI method, and the colour part is the comparison between the proposed method and CBBA. It can be found that the proposed method has the most improvement over CBBA, where the proportion of improved solutions is about 26–90%, with an increase in allocated tasks of 4–8% approximately. Compared with PI, the solution improvement ratio is approximately 15–54%, and an increased task range of approximately 3–7%. In general, the proposed method has an improvement in optimization compared with the other two methods, and the more tasks are presented, the better the improvement effect is, attributed to the enhanced exploration of tasks in the removal stage of the proposed method.

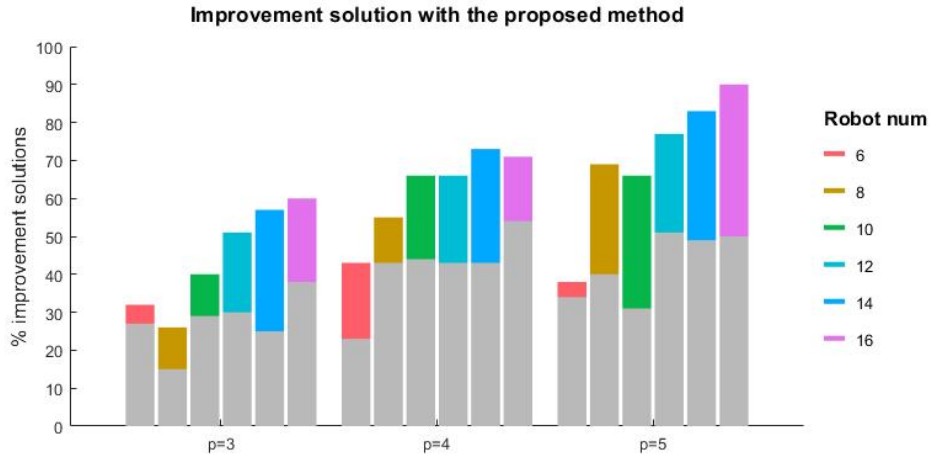

**Figure 9.** The percentage improvement solution with the proposed method over the other two methods in the number of task allocations.

Figure 11 shows the convergence ratio of the three methods in so many tests, and it can be found that both the proposed method and CBBA have complete convergence (100%), while the PI method has partial non-convergence (89–97%). Although the proposed method has the same real-time bid update mechanism as the PI method, which can truly reflect the task cost and better optimize the scheduling, it is easy to cause the algorithm to fall into an endless loop without convergence. Through the endless-loop avoidance mechanism proposed in this paper, the proposed mechanism can monitor the deadlock through historical data and choose the best exit if there exists an infinite loop. Similarly, using this mechanism, the PI method can also fully converge, which verifies the effectiveness of the proposed mechanism.

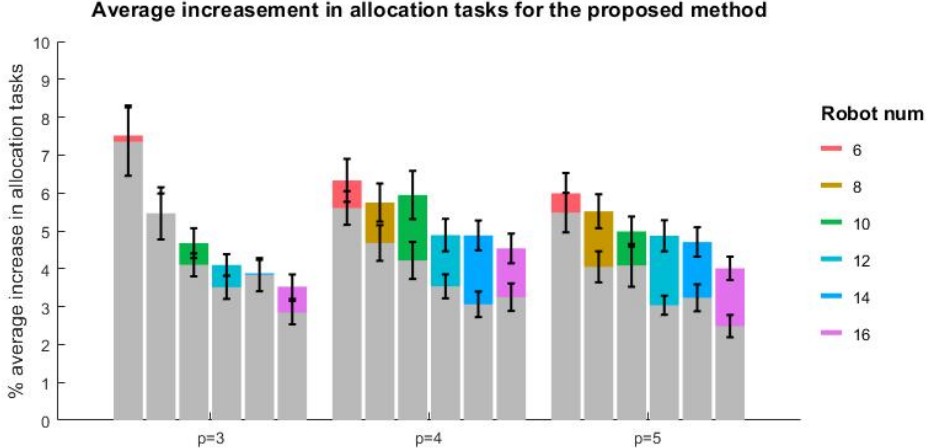

**Figure 10.** The average percentage increase and standard deviation in the number of allocated tasks over the other two methods.

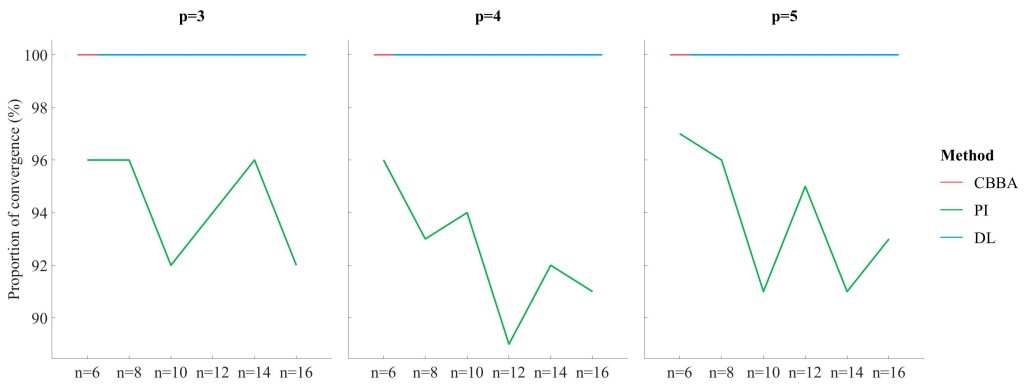

**Figure 11.** The convergence ratio of the three methods.

In summary, the proposed method has a good improvement over the current two most representative methods in the number of iterations and optimization performance, which is attributed to the proposed new frame, new task removal and task inclusion method. Compared with CBBA and PI method, the iterations required by the proposed method are reduced by about 40 and 55%, respectively. The number of allocated tasks is increased by 4–8% and 3–7%, respectively, and the improvement ratio is 26–90% and 15–54%, respectively. Moreover, the proposed method can achieve 100% convergence, which cannot be achieved by the PI method with the same bid update mechanism. This is mainly due to the deadlock avoidance mechanism proposed in this paper, through which it is also able to help PI achieve full convergence.

## 5. Conclusions

In this paper, a double-loop framework is designed to enhance the performance of scheduling, where a new task removal method in the outer loop and a local redundant auction method in the inner loop are proposed to improve the optimization of scheduling and reduce iterations. Furthermore, a deadlock detection mechanism is introduced to avoid endless loops and the scheduling with the lowest local cost will be adopted to exit the cycle. Extensive Monte Carlo experiments show that the iterations required by the proposed method are less than CBBA and PI, and the number of allocated tasks increases. In addition, through the deadlock avoidance mechanism, PI can completely converge as can the method in this paper. Although the proposed method has significantly reduced the amount of communication and computation compared with the centralized method, its

scalability for larger UAV swarms still needs to be further studied in the future. In addition, the security of communication is also worth studying.

**Author Contributions:** Conceptualization, R.C. and J.L.; methodology, R.C.; software, R.C.; validation, J.L., Y.C. and Y.H.; investigation, Y.C. and Y.H.; writing—original draft preparation, R.C.; writing—review and editing, J.L., Y.C. and Y.H. All authors have read and agreed to the published version of the manuscript.

**Funding:** This research received no external funding.

**Data Availability Statement:** Not applicable.

**Acknowledgments:** The authors would like to thank the editors and the reviewers for their most constructive comments to improve the quality of this paper.

**Conflicts of Interest:** The authors declare no conflict of interest.

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
