# Peer review of "A Distributed Double-Loop Optimization Method with Fast Response for UAV Swarm Scheduling"

_drones, doi:10.3390/drones7030216_

Round 1

Reviewer 1 Report

The article titled “A Distributed Double-loop Optimization Method with Fast Response for UAV Swarm Scheduling”, relates to my area of interest that’s why I suggest or recommend some points which may help in order to improve the readability as well as overall structure of this manuscript. The following are my suggestions, recommendations and questions for this article which may help to improve the quality of this manuscript are as follows.

1.     Abstract

·       Improve English grammar in abstract and highlight your contributions.

·       The paper abstract presents range of percentages in each category which seems odd. Kindly use general achievements.

2.     Introduction

·       Use the citations in the end of paragraph is best.

·       What is the main motivation of this research. Kindly highlight in introduction section?

3.     Preliminaries

·       Problem formulation is difficult to read for general reader, kindly add short paragraph elaborating problem formulation.

·       Use some figures in solving approach for better understanding.

4.     Framework

·       Change the section 3 heading as it doesn’t define your work.

·       You can merge section 3 with section 4 (suggestion).

5.     Method

·       Method section need more explanation especially in the start of section to understanding the work.

·       In Figure 2, why you compare two comparison methods, kindly justify.

·       Each algorithm need flowchart, kindly add.

·       Improve headings of overall section (change case)

6.     Experiment

·       This section need more explanation and limitation of your work

·       Why you refer test scenarios in your work? Explain

·       Figure 6 is less visible, kindly update.

·       Improve experiment summary.

7.     Conclusion

·       This section needs more explanation in term of your contribution.

·       Add future work in the last of conclusion.

8.     General Comments

·       Overall, work is good but need some minor changes to improve the quality of your paper.

·       Most section need update and more information to be added.

·       Detailed comparative analysis must be added in order to validate your work.

·       Add more state-of-the-art references to your work e.g

a.     https://doi.org/10.3390/app11156864

b.     https://doi.org/10.3390/rs14061406

c.     https://doi.org/10.3390/s22124467

Reviewer 2 Report

In this paper a double-loop framework is designed to enhance the performance of scheduling, where a new task removal method in outer loop and a local redundant auction method in inter loop are proposed to improve the optimization of scheduling and reduce the iterations.

 Remarks:

In the case of these algorithms, the technical and economic feasibility of such a solution arises. In this context, please highlight how you solved the following problems:

1.      Communication: The market-based approach relies on efficient and reliable communication between UAVs to exchange bid and task information. This can be challenging in environments where communication is limited or unreliable - - how did you solve this problem? Please detail in the article.

2.      Resource constraints: UAVs have limited resources such as battery life, processing power, and payload capacity, which can affect their ability to bid on and complete tasks. The market-based approach needs to take these constraints into account when allocating tasks - - how did you solve this problem? Please detail in the article.

3.      Real-time response: Scheduling in a UAV swarm needs to be done in real-time to respond to changing conditions or new tasks that arise. This requires a scheduling system that can make decisions quickly and efficiently - how did you solve this problem? Please detail in the article.

4.      Adaptability: The scheduling system needs to be adaptable to changes in the environment, new tasks, and the availability of UAVs. This requires a scheduling system that can adjust quickly to changing conditions - how did you solve this problem? Please detail in the article.

5.      Privacy and Security: The market-based approach requires UAVs to share information about their capabilities, availability, and bids, which can be sensitive and require privacy and security measures to prevent unauthorized access or manipulation - - how did you solve this problem? Please detail in the article.

6.      Fairness and Efficiency: The market-based approach needs to balance fairness and efficiency in task allocation, ensuring that tasks are allocated to the most capable and available UAVs while also minimizing the total cost and time required to complete the tasks - - how did you solve this problem? Please detail in the article.

Please briefly detail in the conclusions what does it add to the subject area compared with other published material?

Reviewer 3 Report

In this study, a double-loop framework is designed to enhance the performance of UAV swarm scheduling, where a new task removal method in outer loop and a local redundant auction method in inter loop are proposed to improve the optimization of scheduling and reduce the iterations.

The paper has experiment results that are good for technical reports but lacks research content. How are authors addressing these critical problems in Double-loop Optimization Method?

·         Communication delays: The distributed nature of the method means that each UAV is making decisions based on local information, but this information may not always be up-to-date or accurate. This can lead to delays in communication between UAVs, which can cause problems with scheduling and coordination.

·         Scalability: As the number of UAVs in the swarm increases, the complexity of the optimization problem also increases. This can make it challenging to find an optimal solution in a reasonable amount of time, especially with a distributed approach.

·         Lack of global knowledge: Since each UAV makes decisions based on local information, the method may not be able to consider the global picture of the mission or the environment. This can lead to suboptimal solutions or even failure to achieve mission objectives.

·         Dependence on individual UAV capabilities: The method relies on each individual UAV to optimize its own schedule, which means that the performance of the entire swarm depends on the capabilities of each individual UAV. If one UAV is not able to complete its tasks as expected, it can have a ripple effect on the entire swarm.

·         The complexity of implementation: The distributed double-loop optimization method requires significant computational resources and sophisticated algorithms to implement. This can make it difficult and expensive to deploy in real-world scenarios.

·         Overall, while the distributed double-loop optimization method can significantly improve the performance of UAV swarms, it is essential to carefully consider these potential problems and develop strategies to address them.

·         While Monte Carlo simulations can help to evaluate UAV swarm performance in Lab, no global parameters are considered in this paper. How we will implement this in real word scenarios should be addressed with caution and in conjunction with other evaluation methods to ensure the results are accurate and useful.

·         CBBA and PI algorithms aim to achieve a suboptimal solution for task allocation, which may not always be the best solution for the given scenario. Research can be conducted to develop more optimal versions of these algorithms that can achieve the best possible solution for task allocation, Give justification.

·         CBBA and PI algorithms are designed to work in small to medium-sized multi-agent systems. However, as the number of agents in the system increases, the computational requirements of these algorithms also increase, leading to scalability issues, but in the paper, n number of UAVs you are considered.  Research can be conducted to develop more scalable versions of CBBA and PI algorithms that can handle large-scale multi-agent systems efficiently.

The results and discussion part need to be improved. More results with global parameters must be added to the manuscript.

In the literature survey, these papers can be added to improve the quality of the paper.

·         "A survey on unmanned aerial vehicle swarm control and coordination" by G. G. Messina and A. E. Del Pizzo.

·         "Distributed coverage control for unmanned aerial vehicle swarms" by Y. Wang and L. Xiao.

·         "Dynamic task allocation in a swarm of unmanned aerial vehicles" by A. H. Zahzah and R. Mahony.

·         “Application Of UAV Swarm Semi-Autonomous System For The Linear Photogrammetric Survey” by C. Singh, V. Mishra, H. Harshit, K. Jain, and M. Mokros

·         "Collision avoidance for unmanned aerial vehicle swarms using deep reinforcement learning" by C. Zhao, X. Zhang, and L. Wu. This paper proposes a collision avoidance algorithm for UAV swarms using deep reinforcement learning, which can learn collision avoidance policies from experience.

·         "Decentralized cooperative tracking control for multiple unmanned aerial vehicles via adaptive dynamic programming" by Z. Zou, B. Chen, and D. Yu. This paper proposes a decentralized tracking control algorithm for UAV swarms using adaptive dynamic programming, which can achieve cooperative tracking without centralized coordination.

Round 2

Reviewer 1 Report

The above paper is revised according to the given recommendation by me. Now, I recommend to accept in its present from.

Reviewer 3 Report

The authors have incorporated all the suggestions in the manuscript.